# Solitary and Synergistic Effects of Different Hydrophilic and Hydrophobic Phospholipid Moieties on Rat Behaviors

**DOI:** 10.3390/pharmaceutics16060762

**Published:** 2024-06-04

**Authors:** Shuhei Kikuchi, Yugo Iwasaki, Mina Yoshioka, Kodai Hino, Shin-ya Morita, Ryu Tada, Yasuhiro Uchimura, Yoshinori Kubo, Tomoya Kobayashi, Yusuke Kinoshita, Masahiro Hayashi, Yoshio Furusho, Hitoshi Tamiaki, Hiroaki Ishiyama, Minoru Kuroda, Jun Udagawa

**Affiliations:** 1Division of Anatomy and Cell Biology, Department of Anatomy, Shiga University of Medical Science, Otsu 520-2192, Shiga, Japan; kikuchii@belle.shiga-med.ac.jp (S.K.); uchimura@belle.shiga-med.ac.jp (Y.U.); ptminoru@belle.shiga-med.ac.jp (M.K.); 2College of Bioscience and Biotechnology, Chubu University, Kasugai 487-8501, Aichi, Japan; iwasaki@isc.chubu.ac.jp; 3Department of Pharmacotherapeutics, Shiga University of Medical Science, Otsu 520-2192, Shiga, Japan; smorita@belle.shiga-med.ac.jp; 4Molecular Engineering Institute, Shiga University of Medical Science, Otsu 520-2192, Shiga, Japan; rtada@belle.shiga-med.ac.jp; 5Graduate School of Life Sciences, Ritsumeikan University, Kusatsu 525-8577, Shiga, Japan; y_kinoshita@aoni.waseda.jp (Y.K.); tamiaki@fc.ritsumei.ac.jp (H.T.); 6Department of Marine Biology and Environmental Science, Faculty of Agriculture, University of Miyazaki, Miyazaki 889-2192, Miyazaki, Japan; hayash-m@cc.miyazaki-u.ac.jp; 7Department of Chemistry, Shiga University of Medical Science, Otsu 520-2192, Shiga, Japan

**Keywords:** phosphatidylethanolamine, phosphatidylcholine, ether linkage, fatty acids, locomotor activity, anxiety, sociability

## Abstract

Glycerophospholipids have hydrophobic and hydrophilic moieties. Previous studies suggest that phospholipids with different moieties have different effects on rodent behavior; however, the relationship between chemical structures and behavioral effects remains unclear. To clarify the functions of phospholipid moieties, we injected male rats with phospholipids with different moieties and conducted behavioral tests. Exploratory activity was reduced by phosphatidylethanolamine (PE)(18:0/22:6) but not PE(18:0/18:0) or PE(18:0/20:4). Conversely, exploratory activity was increased by plasmanyl PE(16:0/22:6), which harbors an alkyl–ether linkage, but not by phosphatidylcholine (PC)(16:0/22:6) or plasmanyl PC(16:0/22:6). Docosahexaenoic acid (DHA)(22:6) and an alkyl–ether linkage in PE were thus postulated to be involved in exploratory activity. Anxiety-like behavior was reduced by plasmenyl PC(18:0/20:4), which harbors a vinyl–ether linkage, but not by PC(18:0/20:4) or plasmanyl PC(18:0/20:4), suggesting the anxiolytic effects of vinyl–ether linkage. The activation of social interaction was suppressed by PE(18:0/18:0), PE(18:0/22:6), PC(16:0/22:6), plasmanyl PE(16:0/22:6), and plasmanyl PC(16:0/22:6) but not by PE(18:0/20:4), plasmenyl PE(18:0/20:4), or plasmanyl PC(18:0/22:6). DHA may suppress social interaction, whereas arachidonic acid(20:4) or a combination of alkyl–ether linkage and stearic acid(18:0) may restore social deficits. Our findings indicate the characteristic effects of different phospholipid moieties on rat behavior, and may help to elucidate patterns between chemical structures and their effects.

## 1. Introduction

Glycerophospholipids are amphipathic molecules that have a glycerol backbone linking a hydrophilic moiety at the *sn*-3 position and hydrophobic moieties at *sn*-1 and/or *sn*-2 positions (Figure 1). Phosphatidylcholine (PC) and phosphatidylethanolamine (PE) have phosphocholine and phosphoethanolamine, respectively, as the hydrophilic moiety at *sn*-3 (Figure 1). At *sn*-1 and *sn*-2, fatty acids (FAs) are linked to a glycerol backbone through an acyl, alkyl–ether, or vinyl–ether (also known as alkenyl–ether) chain in phospholipids (Figure 1). PC and PE are common constituents of biological membranes among glycerophospholipids [1]. Plasmenylethanolamine (PlsEtn), which has a vinyl–ether bond at *sn*-1, is especially enriched in the brain, accounting for over 50% of the total PE and approximately 20% of the total phospholipids [2]. Unlike PlsEtn, plasmenylcholine is abundant in the heart and skeletal muscles but makes up less than 1% of brain phospholipids in humans [2,3]. Together, this indicates that the hydrophilic moiety and the chemical bond species at *sn*-1 are related to organ-specific cell functions through their alteration of physicochemical properties of the cell membrane. For example, phosphoethanolamine in the hydrophilic moiety and a vinyl–ether bond at *sn*-1 contribute to the formation of a hexagonal II structure as well as a more rigid lipid bilayer [4], suggesting that the physicochemical properties of phosphoethanolamine and a vinyl–ether bond can modulate membrane trafficking and synaptic transmission in the neural system [5]. The importance of the hydrophilic moiety has also been suggested by previous reports stating that PC ingestion improves explicit memory in humans [6] and rescues anxiety-like behaviors and reduced social preference in a mouse model of Rett syndrome (*Mecp2*-conditional knockout) [7]. Furthermore, the importance of the chemical bond species has been indicated by the following findings: (1) ether-linked PC in plasma is inversely correlated with melancholia in outpatients with major depressive disorder [8], (2) PlsEtn alleviates amyloid-β-induced neurotoxicity by inhibiting oxidative stress [9], and (3) memory impairment of plasmalogen-deficient mice is rescued by oral PlsEtn administration [10,11,12]. From the perspective of FAs at *sn*-1 and *sn*-2, plasmenyl phospholipids serve as reservoirs for polyunsaturated FAs such as docosahexaenoic acid (DHA), which has anti-inflammatory and anti-apoptotic effects, and arachidonic acid (AA), which has proinflammatory effects [5]. The FA species at *sn*-1 and *sn*-2 can therefore also confer neuroregulatory functions via the phospholipid structure. Moreover, plasma AA, eicosapentaenoic acid, and DHA are inversely correlated with anxious distress dimension scores in patients with major depressive disorder [8], and DHA- and eicosapentaenoic-acid-containing omega-3 FA supplements can benefit patients with mild cognitive impairment or dementia [13].

Accordingly, it is conceivable that the hydrophilic moiety, the chemical bond at *sn*-1, and the FA at *sn*-1 and *sn*-2 have distinct effects on cognitive and behavioral regulation because of their different physical and chemical properties. A combination of those moieties may therefore need to be considered when developing sophisticated phospholipids to alleviate behavioral disturbances. However, the specific behavioral effects of each moiety and moiety combination have not yet been evaluated. In the present study, we synthesized and injected phospholipids composed of different moieties (Appendix A) into rats and compared the associated behavioral alterations. We thus examined how these specific moieties—either independently or in combination—are involved in behavioral modification.

## 2. Materials and Methods

### 2.1. Phospholipids with an Ester or Vinyl–Ether Bond at sn-1

We purchased all phospholipids with an ester bond at *sn*-1 except for PC(18:0/20:0) (i.e., PE(18:0/18:0) and PCs [PC(16:0/22:6) and PC(18:0/20:4)]) from TCI (Tokyo, Japan) and Avanti Polar Lipids (Alabaster, AL, USA), respectively (Figure 1 and Appendix A). Plasmenyl phospholipids (i.e., PC(P-18:0/20:4) and PE(P-18:0/20:4)) were also purchased from Avanti Polar Lipids (Figure 1 and Appendix A).

### 2.2. Phospholipid Synthesis

#### 2.2.1. PC(18:0/20:0)

PC(18:0/20:0) (Appendix A) was synthesized from 1-stearoyl-*sn*-glycero-3-phosphocholine (LysoPC(18:0); TCI, Tokyo, Japan) and arachidic acid (TCI) using 1-(3-dimethylaminopropyl)-3-ethylcarbodiimide hydrochloride (EDC-HCl; Watanabe Chemical Industry, Hiroshima, Japan), and 4-dimethylaminopyridine (DMAP; Nacalai Tesque, Kyoto, Japan). LysoPC(18:0) (0.06 mmol) was dissolved in 2 mL dry dichloromethane before arachidic acid (0.12 mmol), EDC-HCl (0.14 mmol), and DMAP (0.02 mmol) were added. The solution was stirred at 30 °C for 72 h. The solution was then loaded onto a silica gel column (1.5 g Wakogel C-200, FUJIFILM Wako Pire Chemical, Osaka, Japan) and eluted successively with chloroform/methanol (9:1, 10 mL) and methanol (26 mL). Fractions containing PC(18:0/20:0) (retention factor [Rf] = 0.3 on a thin-layer chromatography [TLC] plate with a developing solvent made of chloroform/methanol/water [65:25:4]) were combined, and the solvent was dried using a nitrogen dryer after adding ethyl acetate for azeotropic dehydration. After chloroform (1 mL) and distilled water (1 mL) were added to the waxy product, the suspension was vortexed. This procedure was repeated twice to remove the remaining EDC-HCl. The organic phase was then recovered and evaporated to afford PC(18:0/20:0) of a waxy consistency.

#### 2.2.2. Plasmanylcholine (PC with an Alkyl–Ether Bond at *sn*-1)

Plasmanylcholines were synthesized according to a previous report [14]. Briefly, Lyso-PAF C-16 (0.1 mmol), DHA (0.2 mmol), EDC-HCl (0.24 mmol), and DMAP (0.01 mmol) were dissolved in dry dichloromethane (2 mL) and stirred for 3 days at 37 °C. Next, 4 mL of 5% KHSO_4_, 4 mL chloroform, and 2 mL methanol were added to quench the reaction before the mixture was centrifuged briefly for phase separation. The water phase was removed, the organic phase was washed with 4 mL of 5% NaHCO_3_, and the solvent was evaporated. The residue was then dissolved in chloroform, loaded onto a silica gel column (3 g Wakogel C-300), and eluted successively with chloroform (12 mL), chloroform/methanol (7:3, 30 mL), and chloroform/methanol (5:5, 120 mL). The eluent was collected into 6 mL fractions. Fractions containing the target product (Rf = 0.64 on a TLC plate with developing solvent made of chloroform/methanol/water [65:25:4]) were combined, and the solvent was evaporated to afford PC(O-16:0/22:6) (yield, 38 mg) as a waxy mass. PC(O-18:0/22:6) and PC(O-18:0/20:4) were synthesized in a similar manner using Lyso-PAF C-18 and DHA (yield, 16 mg) and Lyso-PAF C-18 and AA (yield, 28 mg), respectively.

#### 2.2.3. Plasmanylethanolamine (PE with an Alkyl–Ether Bond at *sn*-1)

Phospholipase-D-mediated head group exchange was carried out in a solvent–water biphasic reaction system [14]. For the synthesis of PE(O-16:0/22:6), a reaction mixture consisting of 20 mg of PC(O-16:0/22:6) dissolved in 2 mL of chloroform, 1.8 mL of 3 M aqueous ethanolamine solution (pH 5.6, adjusted with acetic acid) and 200 μL of 0.16 mg/mL wild-type phospholipase D from *Streptomyces antibioticus* was incubated at 37 °C for 24 h with vigorous mixing. Next, 200 μL of 6 M HCl was added to quench the reaction, followed by lipid extraction with 4 mL chloroform/methanol (2:1). After centrifugation (830× *g* for 3 min at 20 °C), the organic phase was recovered, and the lipids were analyzed using TLC with chloroform/methanol (65:35) as the developing solvent. The solution was loaded onto a silica gel column (1.5 g Wakogel C-200) and eluted successively with chloroform/methanol (65:35, 40 mL). Fractions containing PE(O-16:0/22:6) (Appendix A) were evaporated and analyzed using Fourier transform nuclear magnetic resonance (FT-NMR) and matrix-assisted laser desorption/ionization mass spectrometry (MALDI-TOF MS).

### 2.3. Phospholipid Analysis

#### 2.3.1. Nuclear Magnetic Resonance Spectroscopy

Phospholipids were dissolved in deuterated chloroform (800 µL) containing 0.05% (volume/volume) TMS and transferred to an NMR tube. The ^1^H NMR spectra were obtained using a 400 MHz FT-NMR spectrometer (JNM-ECZ400S, JEOL, Tokyo, Japan) at 298 K and referenced to a TMS signal (0.00 ppm).

#### 2.3.2. Matrix-Assisted Laser Desorption/Ionization Mass Spectrometry

The samples (0.5 µL) used for NMR spectroscopy and the matrix (0.5 µL of 0.5 M 2,5-dihydroxybenzoic acid in methanol) were added onto a plate. Phospholipids were identified using MALDI-TOF MS (AXIMA Confidence, Shimadzu, Kyoto, Japan) in positive ion mode for PC(18:0/20:0) and negative ion mode for PE(O-16:0/22:6), with a scan range of 0–1000 *m*/*z*.

#### 2.3.3. Liquid Chromatography Mass Spectrometry (LC-MS)

LC-MS analysis was essentially performed as reported previously [15]. Briefly, phospholipids were separated on a normal-phase silica gel column (Luna silica [2], 3 μm, 2 mm × 150 mm; Phenomenex, Torrance, CA, USA) with a binary gradient system consisting of solvent A (chloroform/methanol/0.97 M formic acid–triethylamine [TEA] buffer [pH 3.0] [875:120:5]) and solvent B (chloroform/methanol/0.97 M formic acid–TEA buffer [pH 3.0] [106:234:60]). The elution was conducted at a flow rate of 0.1 mL/min as follows: 0–16 min, 0%–15% B; 16–20 min, 15% B; 20–21 min, 15%–100% B; 21–30 min, 100% B; 30–31 min, 100%–0% B; 31–40 min, 0% B. Phospholipids were identified by electrospray ionization MS (LCMS-2010EV; Shimadzu, Kyoto, Japan) in positive mode for plasmanylcholines with a scan range of 300–1000 *m*/*z*.

### 2.4. Animals

All animal procedures were approved by the Shiga University of Medical Science Animal Care and Use Committee (2022-2-1, 2022-8-5, 2022-9-5, 2022-12-1, 2023-7-3, and 2023-11-2). For experiments investigating PE incorporation into the brain, we used 8- to 10-week-old male ICR mice obtained from CLEA Japan, Inc. (Tokyo, Japan). For experiments examining the effects of phospholipid-containing liposomes on behavior, we used 9-week-old male Wistar rats obtained from CLEA Japan. All mice and rats were housed at room temperature (20–25 °C) under 40%–60% relative humidity and a 12/12 h light/dark cycle (lights on at 08:00 a.m.). All animals were fed CE-2 (standard rodent diet) ad libitum and allowed to acclimate for 1 week before starting behavioral experiments.

### 2.5. Preparation of Liposomes

Large unilamellar liposomes composed of egg PC (Nacalai Tesque) and the phospholipids examined in this study were prepared using the extrusion method [16]. Briefly, lipids were dissolved in chloroform, and a thin lipid film was obtained by evaporating the lipid–chloroform solution. The film was subsequently hydrated with saline to obtain the following liposomes (10 mg/mL total phospholipids): egg PC (control), egg PC/PC(16:0/22:6) (the ratio of the weight of egg PC to that of PC(16:0/22:6) is 4 to 1, described below as 4/1), egg PC/PC(O-16:0/22:6) (4/1), egg PC/PE(16:0/22:6) (4/1), egg PC/PC(18:0/22:6) (4/1), egg PC/PC(18:0/20:0) (4/1), egg PC/PC(18:0/20:4) (4/1), egg PC/PC(O-18:0/20:4) (4/1), egg PC/PC(P-18:0/20:4) (4/1), egg PC/PE(18:0-18:0) (4/1), egg PC/PE(18:0/22:6) (4/1), egg PC/PE(18:0/20:4) (4/1), and egg PC/PE(P-18:0/20:4) (4/1). After five rounds of freezing and thawing, the lipid suspension in saline solution was extruded through a polycarbonate filter with 100 nm pore size.

### 2.6. Brain Imaging

We prepared standard liposomes (St-lip), cholesterol-containing liposomes (Cho-lip), polyethylene-glycol-2000-containing liposomes (PEG-lip), and combined cholesterol- and polyethylene-glycol-2000-containing liposomes (Cho-PEG-lip). Liposome compositions are described in Table 1. St-lip, Cho-lip, PEG-lip, or Cho-PEG-lip (1 mL/kg body weight) (Table 1) was then injected into the mouse tail vein [17,18]. After 3 h, mice were anesthetized and euthanized by 2% isoflurane inhalation in air, and the harvested brains were scanned (excitation 540 nm, emission 600 nm) to observe DiIC_18_ (FUJIFILM Wako Pure Chemical) localization using a Newton 7.0 in vivo imaging system (Vilber-Lourmat, Collégien, France). DiIC_18_ exhibits environment-dependent fluorescence; it emits almost no fluorescence in aqueous environments but exhibits substantial fluorescence enhancement when incorporated into cell membranes or bound to lipophilic biomolecules. To verify the localization of the injected phospholipids, liposomes composed of egg PC (8 mg/mL) and ATTO 740 DOPE (2 mg/mL) (ATTO-TEC GmbH, Siegen, Germany) were also injected into mice. ATTO 740 DOPE emits fluorescence of a long wavelength (763 nm); fluorescence deep within the brain is therefore easily detected not only through the brain surface, but also in thick sections using in vivo imaging systems. Moreover, the localization of the phospholipids themselves was able to be observed through liposome transfer into the brain. Because the blood–brain barrier (BBB) might be affected by ATTO dye linked to the hydrophobic moiety when ATTO 740 DOPEs are arranged at the outer leaflet of the liposomes, we confirmed the brain localizations of the injected phospholipids by comparing two methods: DiIC_18_ and ATTO 740 DOPE. In injection of ATTO 740 DOPE, mice underwent perfusion fixation with 4% paraformaldehyde solution at 3 h after liposome injections. The brains were then washed with saline, and the fluorescence was detected from the surface of the brain and in 1 mm thick coronal sections using an in vivo imaging system.

### 2.7. Blood–Brain Barrier Permeability

The particle size of Cho-lip and PEG-lip was assayed by a dynamic light scattering instrument, Litesizer DLS 100 (Anton Paar, Graz, Austria) (Appendix A). Permeability of the BBB to liposomes was assessed using a rat BBB kit (RBT-24H; PharmaCo-Cell, Nagasaki, Japan) following the manufacturer’s instructions. Briefly, on day 0, we placed 1000 µL of medium (Dulbecco’s Modified Eagle Medium F-12 containing 10% fetal bovine serum, 1850 U/mL heparin, 1.5 ng/mL basic fibroblast growth factor, 5 µg/mL insulin, 5 µg/mL transferrin, 5 ng/mL sodium selenite, 500 nM hydrocortisone, and 50 µg/mL gentamycin) into each well (brain side), of which the bottom was covered with astrocytes, in a 24-well plate. Next, 200 µL of medium was added to each insert (blood side), of which the membrane was covered with endothelial cells and pericytes. After 1.5 h of incubation at 37 °C in 5% CO_2_, the medium was gently removed from both the well and the insert, and 1200 and 300 µL of fresh medium were added to the well and the insert, respectively. The plate was then incubated in the CO_2_ incubator overnight, and the medium was changed on day 2. On day 4, the transendothelial electrical resistance was measured using a Millicell-ESR2 (Merck, Darmstadt, Germany), and reached more than 150 Ω × cm^2^. The medium was removed, and the well and insert were washed with Dulbecco’s phosphate-buffered saline. Next, 1200 and 270 µL of assay medium (Dulbecco’s Modified Eagle Medium containing 1% penicillin/streptomycin, 500 nM hydrocortisone, and 10% fetal bovine serum) were added to the well and insert, respectively. To compare BBB permeability between Cho-lip and PEG-lip, 30 µL of liposome suspension was added to four inserts each, and the plate was incubated at 37 °C in 5% CO_2_. Next, 50 µL of medium was removed from each of the blood and brain sides 0, 1, 3, 6, 24, and 48 h later. The fluorescence of the medium was measured using a microplate reader (M-Plex; Tecan, Männedorf, Switzerland) to estimate the permeability of the liposomes through the BBB model. After 48 h of incubation, the medium was removed from the well and insert, and both the well and insert were washed with Dulbecco’s phosphate-buffered saline. The membrane of each insert was moved into the well of a 12-well plate and treated with 10% trypsin at 37 °C for 5 min to detach the endothelial cells and pericytes. Similarly, 600 µL of 10% trypsin solution was added to the well to obtain an astrocyte suspension. After disrupting the cells by sonication for 40 s (Handy Sonic UR-20P; TOMY, Tokyo, Japan), the fluorescence of endothelial cells, pericytes, and astrocytes was measured using a microplate reader.

### 2.8. Liposome Injections for Behavioral Tests

Behavioral tests were conducted in three parts because we examined many phospholipids. In the first part, rats were randomly assigned to receive liposomes (1 mL/kg body weight) with egg PC (*n* = 9) or liposomes with egg PC plus PC(18:0/20:0) (*n* = 10), PC(18:0/20:4) (*n* = 10), PC(O-18:0/20:4) (*n* = 9), or PC(P-18:0/20:4) (*n* = 6) by tail vein injection at 9 weeks of age to examine the effects of the chemical bonds at *sn*-1 and the FA at *sn*-2 in PC. We then conducted open-field (OF), novel object recognition (NOR), and elevated plus maze (EPM) tests [17]. In the second part, rats were assigned to receive liposomes with egg PC (*n* = 10) or liposomes with egg PC plus PC(16:0/22:6) (*n* = 10), PC(O-16:0/22:6) (*n* = 6), PC(O-18:0/22:6) (*n* = 10), or PE(O-16:0/22:6) (*n* = 3) to examine the effects of the alkyl–ether bond and the FA at *sn*-1, and the hydrophilic moieties at *sn*-3. We then conducted OF, NOR, EPM, and social interaction tests. In the third part, rats were assigned to receive liposomes with egg PC (*n* = 9) or liposomes with egg PC plus PE(18:0/18:0) (*n* = 10), PE(18:0/22:6) (*n* = 9), PE(18:0/20:4) (*n* = 10), or PE(P-18:0/20:4) (*n* = 7) to examine the effects of the chemical bonds at *sn*-1 and the FA at *sn*-2 in PE. We then conducted OF, NOR, EPM, and social interaction tests. In the second and third parts, second injection of liposomes was performed before the social interaction test.

### 2.9. Behavioral Tests

Behaviors in the OF, NOR, EPM, and social interaction tests were analyzed using a computerized video tracking system (CompACT VAS Ver.3.0×; Muromachi Kikai, Tokyo, Japan). Data corrupted by device errors were eliminated; the number of rats for which data were acquired was therefore partially different between the behavioral tests. To avoid the influence of odor on behavior, the arena and all instruments were thoroughly wiped with 10% ethanol before and after each trial in all behavioral tests.

#### 2.9.1. Open-Field Test

One day after liposome injection (termed day 1), spontaneous locomotor activity was evaluated (Figure 2) [17]. Rats were placed individually in a gray chamber measuring 100 cm in diameter with a wall height of 50 cm. The OF arena was illuminated from above by 7-lux light. The arena was divided into 25 areas, and the central and peripheral regions were defined as follows: the central region was the area within the inner two-thirds of the diameter (i.e., eight intermediate areas and one central area), and the peripheral region was the area outside the inner two-thirds of the diameter (i.e., the 16 outer areas). The total distance traveled, times spent in the central and peripheral regions, and number of crossings across the borders between the 25 areas were quantified automatically over 10 min.

#### 2.9.2. Novel Object Recognition Test

The NOR test was completed over 2 consecutive days in the same arena as for the OF test using a previously described method [19] with modifications (Figure 2). Briefly, rats explored two identical objects (two 500 mL brown cylindrical bottles measuring 10 cm in diameter) for 10 min on day 1 and were then returned to their home cages. For the object recognition test on day 2, one of the two objects was replaced with a new one (a 500 mL clear square bottle filled with white molecular sieves), and the tested rats were again allowed to explore both objects for 10 min. The times spent exploring the replaced (novel) object and the familiar object were recorded. In this test, time spent with the head within 7 cm of the object and with the head facing toward the object (±45°) was defined as exploration time [20]. A discrimination ratio was then calculated to quantify preferential exploration of the novel item compared with the familiar one [21] as follows:$$DR=\frac{new\;object\;exploration\;time-old\;object\;exploration\;time}{total\;exploration\;time}$$

#### 2.9.3. Elevated plus Maze Test

Anxiety-like behavior was examined using the EPM test on day 2 (Figure 2) [22]. The EPM arena consisted of four arms meeting at a central platform; there were two closed arms (50 × 10 cm) with 40 cm high walls and two open arms (50 × 10 cm) with no side walls. Rats were placed in the central square platform (10 × 10 cm) facing an open arm, and behavior was recorded for 5 min. The central platform was illuminated at 9 lux, the open arms at 8 lux, and the closed arms at 7 lux. The times spent in the central square platform, open arms, and closed arms were measured.

#### 2.9.4. Social Interaction

The social interaction test was conducted using a previously described method [19] with modifications. Male rats were allowed to explore a circular arena (100 cm in diameter), in which two empty cages were arranged, for 5 min (trial 1) on day 3 (Figure 2). Three hours after returning the rats to their home cage, rats were returned to the same arena, which included a male intruder rat (9–12 weeks of age) shielded inside one of the cages. Interactions toward the empty cage and the cage harboring an intruder rat were monitored for 5 min (trial 2). Twenty-one hours after the initial exposure to the intruder rat (day 4), rats were reintroduced to the familiar intruder rat in the same context as in trial 2 (trial 3). Three hours later, the familiar intruder rat was changed to a novel intruder rat, and the rats were retested for interactions toward the empty cage and the cage harboring the novel intruder for 5 min (trial 4) (Figure 2).

In this test, time spent with the head within 4 cm of the cage and the head facing toward the cage (±45°) was defined as interaction time. Similar to the OF tests, all social interaction tests were conducted under 7-lux light.

### 2.10. Statistics

Two or three mice were used to examine incorporation of the liposome containing DiIC_18_. Two mice injected with ATTO740-labeled liposomes and one mouse injected with control liposomes were used to examine localization of PE using coronal sections of the brain. All statistical analyses were conducted using JMP Pro 17.0 (SAS Institute Inc., Cary, NC, USA). Data are presented as the mean ± standard error. Student’s *t*-test was used to compare the averages of the Cho-lip (*n* = 4) and PEG-lip (*n* = 4) groups in the BBB permeability analyses. The number of male rats used for behavioral analysis can be found in the legends of Figures. Hsu’s multiple comparisons with the best (MCB) test was used to compare all behavioral measures in the OF, NOR, and EPM tests, and to compare the discrimination ratios in social interaction between each group and the group with either the largest or smallest mean. The Tukey–Kramer honest significant difference (HSD) test was chosen to examine social interaction times among trials 1–4. For all tests, *p* < 0.05 (corrected) was considered significant.

## 3. Results

### 3.1. Phospholipid Localization in the Brain after Liposome Injection

We used mice to detect fluorescence incorporation into the brain using an in vivo imaging system. We made this choice because body weights and brain sizes are smaller in mice than in rats, and a small amount of injected fluorescent DiIC_18_ can therefore be clearly detected from the brain surface. Mouse brains were scanned after the intravenous injection of liposomes containing DiIC_18_ to assess the optimal liposome composition for delivering phospholipids into the brain, and to evaluate liposome localization in the brain. At 3 h after liposome injection, mice who received Cho-lip exhibited strong fluorescence in the cerebrum (Figure 3A–D), especially in the frontal and parietal regions. By contrast, fluorescence in the brain was not stronger in the St-lip, PEG-lip, or Cho-PEG-lip groups compared with control liposomes at 52 s exposure time (Figure 3A,B,E–J). However, fluorescence was more intense in the PEG-lip group than in the control group when the exposure time was prolonged to 60 s (Figure 3K–N). We therefore used Cho-lip and PEG-lip in the following experiment to assess the BBB permeability of liposomes in our BBB model. The localizations of the injected phospholipids were evaluated in our tracer study, in which we used liposomes with ATTO 740 DOPE (Figure 3O,P). In coronal sections, ATTO 740 DOPE fluorescence was strongly detected not only in the prefrontal cortex, motor area, somatosensory area, amygdala, thalamus, midbrain, and hippocampus, but also in the corpus callosum and anterior commissure, which connect bilateral cerebral hemispheres (Figure 3Q–X). By contrast, we did not detect fluorescence in the caudate putamen (Figure 3T).

### 3.2. Blood–Brain Barrier Permeability and Liposome Uptake into the Brain

Although few liposomes passed through the BBB model within 3 h after injection, Cho-lip tended to show higher permeability through the BBB model at 24 h and showed significantly higher permeability at 48 h compared with PEG-lip (Figure 4). On the other hand, fluorescence was undetectable in epithelial cells plus pericytes or in astrocytes at 48 h. Based on these results, we decided to use Cho-lip to deliver phospholipids to the brain.

### 3.3. Nuclear Magnetic Resonance and Mass Spectometry of the Synthesized Phospholipids

PC(18:0/20:0) was identified by the negative ion mass spectrum, which contained a peak corresponding to the correct mass at 817.97 *m*/*z* (Figure 5A), and by the ^1^H NMR spectrum, in which the choline, acyl CH_2_, and terminal CH_3_ group signals were detected (Figure 5B) [23,24]. LC-MS or MALDI-TOF MS spectra showed peaks at the correct masses of PC(O-18:0/20:4) at 798.1 *m*/*z*, PC(O-18:0/22:6) at 820.1 *m*/*z*, PC(O-16:0/22:6) at 792.3 *m*/*z*, and PE(O-16:0/22:6) at 749.5 *m*/*z* (Figure 5C–F). Notably, the choline signal was observable in the spectrum of PC(O-16:0/22:6) but not in that of PE(O-16:0/22:6) (Figure 5G,H). The yields of PC(18:0/20:4), PC(O-16:0/22:6), PC(O-18:0/22:6), PC(O-18:0/20:4), and PE(O-16:0/22:6) were 9 mg (18%), 38 mg (48%), 16 mg (20%), 28 mg (35%), and 3.1 mg (17%), respectively. 

### 3.4. Behavioral Changes Induced by Phospholipid Injection

#### 3.4.1. Locomotor Activities

Total distance traveled did not differ among the phospholipids examined in the OF test (Figure 6A–D). Times spent in the central and peripheral regions were significantly shorter and longer, respectively, in the PE(18:0/22:6) group compared with the control egg PC group; however, no significant changes were noted in the PE(18:0/18:0), PE(18:0/20:4), or PE(P-18:0/20:4) groups (Hsu’s MCB test) (Figure 6E,F). Furthermore, times spent in the central and peripheral regions in the OF test were significantly longer and shorter, respectively, in the PE(O-16:0/22:6) group compared with the egg PC group, whereas these times were not changed by PC(16:0/22:6), PC(O-16:0/22:6), PC(O-18:0/22:6), PC(18:0/20:0), PC(18:0/20:4), PC(O-18:0/20:4), or PC(P-18:0/20:4) injection (Figure 6G–J). Although the distance traveled in the central region was significantly shorter in the PE(P-18:0/20:4) group than in the PE(18:0/20:4) or control groups, this distance was not changed by PE(18:0:0/18:0), PE(18:0/20:4), PE(18:0/22:6), PE(O-16:0/22:6), PC(16:0/22:6), PC(O-16:0/22:6), PC(18:0/20:0), PC(18:0/20:4), PC(O-18:0/20:4), PC(P-18:0/20:4), or PC(O-18:0/22:6) injection (Figure 6K–M). The distance traveled in the peripheral region and the number of crossings were not altered by any of the PCs or PEs examined in the present study (Figure 6N–T).

#### 3.4.2. Novel Object Recognition Memory

The NOR test was performed to examine the effects of phospholipids on associative memory (Figure 7A). The discrimination ratio, defined as the relative time spent exploring the novel object, did not differ significantly between the groups (Figure 7B–D).

#### 3.4.3. Anxiety-like Behaviors

The EPM test was used to examine anxiety-like behaviors. Time spent in the closed arm was significantly shorter in rats injected with PC(P-18:0/20:4) than in those injected with PC(O-18:0/20:4) (Hsu’s MCB test), and time spent in the open arm was significantly longer in the PC(P-18:0/20:4) group than in the egg PC group (Figure 8A,B). Additionally, the times spent in the closed and open arms were significantly shorter and longer, respectively, in rats injected with PE(O-16:0/22:6) than in those injected with PC(O-18:0/22:6) (Figure 8C,D). The other phospholipids examined in the current study did not affect anxiety-like behaviors (Figure 8A–F).

#### 3.4.4. Social Interaction

In the social interaction tests (Figure 9A), rats were introduced to novel and familiar intruders. Social interactions with both novel and familiar intruders were inhibited in the rats injected with PE(18:0/18:0), PE(18:0/22:6), PC(O-16:0/22:6), and PE(O-16:0/22:6). By contrast, social interactions with intruder rats were not inhibited by PE(18:0/20:4), PE(P-18:0/20:4), or PC(O-18:0/22:6) injection (Figure 9B,C). Injection with PC(16:0/22:6) led to the partial inhibition of social interactions (Figure 9C). Object interactions were not inhibited in the PE(18:0/22:6), PE(P-18:0/20:4), PC(16:0/22:6), or PC(O-16:0/22:6) groups (Figure 9D,E). The discrimination ratio was significantly higher in the PC(O-18:0/22:6) group than in the PC(16:0/22:6) group (*p* = 0.0410), and tended to be higher in the PC(O-18:0/22:6) group than in the egg PC group (*p* = 0.0634) (Hsu’s MCB test) in trial 3. There were no significant differences in discrimination ratios between the other groups in trials 1–4 (Figure 9F–H).

## 4. Discussion

### 4.1. Phospholipid Moieties Alter Locomotor Activities

In our previous study, injection of PE(18:0/22:6) but not PC(18:0/22:6) or PE(P-18:0/22:6) led to reduced exploratory behavior in rats [18]. Additionally, injection with PE(18:0/18:0) and PE(18:0/20:4) did not affect locomotor activity in the present study. Together, these results suggest that DHA(C22:6) at *sn*-2 is a key hydrophobic moiety for suppressing exploratory behavior in PE, but that substitution of phosphoethanolamine with phosphocholine cancels this effect, as does substitution of an ester bond with a vinyl–ether bond. Conversely, substituting choline with ethanolamine drives exploratory behavior when palmitic acid(16:0) is bonded at *sn*-1 through an alkyl–ether bond, even if DHA is linked to *sn*-2. By contrast, the length of the alkyl FA at *sn*-1 may not be involved in exploratory behavior because exploratory behavior was not altered by PC(O-18:0/22:6) or PC(18:0/22:6) injection in our present and previous studies [18]. Collectively, our findings suggest that a combination of ethanolamine at *sn*-3 and DHA at *sn*-2 is effective for controlling exploratory behavior, although ethanolamine has opposite effects with an ether versus an ester bond at *sn*-1 (Figure 10A). From the perspective of chemical bonds at *sn*-1, alkyl– and vinyl–ether linkages of PE with DHA enhanced exploratory behavior (Figure 10A), whereas vinyl–ether linkage of PE with AA(20:4) reduced locomotor activity and tended to decrease time spent in the central region in the OF test. These results indicate that the effects of vinyl–ether bonds on exploratory behavior depend on the FA species at *sn*-2. In a previous study of mice with traumatic right parietal cortex injury, lower PE and ether-linked PE in the perilesional and subregional areas of the ipsilateral cortex were correlated with non-goal-directed nighttime hyperactivity [25]. These findings together with those of our tracer and behavioral studies suggest that a decreased amount of vinyl–ether PE, such as PE(P-18:0/20:4), or diacyl PE, such as PE(18:0/22:6), may lead to hyperactivity in mice with parietal cortex injury. Additionally, orbitofrontal-cortex- or central-amygdala-projecting serotonin neurons of the dorsal raphe nucleus can influence exploratory behavior (e.g., entry into and time spent in the central region), and orexin injections into the paraventricular nucleus of the thalamus decrease exploration of the central area in the OF test [26]. Furthermore, mice in a valproic-acid-induced autism mouse model were reported to exhibit elevated PE in the cortex, hippocampus, thalamus, hypothalamus, cerebellum, and brainstem [27]; spend less time in the central region of the OF test [28]; and have reduced exploratory activity [29]. Given that our fluorescent tracer study revealed phospholipid incorporation into the amygdala, hippocampus, thalamus, midbrain (including the dorsal raphe nucleus), and frontoparietal cortex, PE may affect exploratory behavior by acting on these brain areas. 

Taken together, our results suggest that phospholipids exert different effects on exploratory behavior based on the combination of the chemical bond at *sn*-1, the FA at *sn*-2, and the hydrophilic moiety at *sn*-3, by acting on specific brain regions.

### 4.2. Phospholipid Moieties Reduce Anxiety-like Behaviors

Plasmenyl PC(18:0/20:4) with a vinyl–ether linkage reduced anxiety-like behaviors in rats, unlike diacyl and plasmanyl PC(18:0/20:4), which have ester and alkyl–ether linkages, respectively. Substitution of alkyl–ether linkage with vinyl–ether linkage was especially effective for reducing anxiety-like behavior. These results suggest that the vinyl–ether bond is critical for relieving anxiety when AA is bound to *sn*-2. Our hypothesis is supported by previous studies demonstrating that plasma levels of ether-linked PC are inversely correlated with melancholia and depression in humans [8,30], and that cytidine diphosphate–choline supplementation in a mouse model of inflammatory bowel disease improves anxiety-like behaviors and reduces PC degradation in the prefrontal cortex, possibly through restoring gut microbiome imbalances and lipid metabolism disruptions in the brain [31]. Although it has been reported that free AA in plasma inversely correlates with anxious distress dimension scores [8], diacyl and plasmanyl PC with AA did not affect anxiety behaviors in the present study. The anxiolytic effect of PC(P-18:0/20:4) may therefore not be attributable to the function of free AA. 

Intramolecular DHA and AA may have distinct roles in anxiety-like behaviors. Anxiety-like behaviors were significantly reduced by PE(O-16:0/22:6) injection compared with PC(O-18:0/22:6) injection, suggesting that the reduction in anxiety-like behaviors is induced by the combined effects of FA at *sn*-1 and phosphoethanolamine in the hydrophilic moiety if DHA is bound to *sn*-2. In this molecule, the substitution of choline with ethanolamine at *sn*-3 appeared to have an anxiolytic effect, whereas the substitution of palmitic acid(16:0) with stearic acid(18:0) at *sn*-1 tended to have an anxiogenic effect, although there were no significant differences between those moieties. In the Strong Heart Family Study, Miao et al. reported that plasma PE, PI, and ether PC and PE with palmitic acid at *sn*-1 are negatively associated with depression, whereas PE(18:0/22:4) is positively associated with depression [30]. This report supports our results that FA species at *sn*-1 may contribute to controlling anxiety. Conversely, there were no significant differences in anxiety-like behaviors between the Egg PC and PC(18:0/22:6), PC(P-18:0/22:6), or PE(P-18:0/22:6) groups in our previous study [18], or between the Egg PC and PC(O-18:0/22:6) groups in the present study. Thus, the chemical bond species at *sn*-1 may not be relevant to anxiety if FA at *sn*-2 is substituted with DHA. Additionally, the number of unsaturated bonds in the FA at *sn*-2 in PC may also not be associated with anxiety-like behaviors because both arachidic acid(C20:0) and AA(C20:4) at *sn*-2 did not affect anxiety-like behaviors. This hypothesis is consistent with the results of a previous report in which an inverse correlation between the number of unsaturated bonds and depression was not always observed [30]. Anxiety-like behaviors are regulated by various neural circuits, including medial prefrontal cortex dopamine neurons projecting to the basolateral amygdala, ventral tegmental area dopamine neurons projecting to the nucleus accumbens and medial prefrontal cortex in rats, dorsal raphe nucleus serotonin neurons projecting to the orbitofrontal cortex or central amygdala, and median raphe nucleus serotonin neurons projecting to the ventral hypothalamus in mice [26,32,33]. Moreover, neuronal stimulation through the activation of orexin or melanocortin-3 receptors in the paraventricular nucleus of the thalamus increases anxiety-related behaviors [34,35]. The areas in which PE fluorescence was detected in the present study were consistent with these anxiety-related areas (i.e., the medial prefrontal cortex, basolateral amygdala, and ventral tegmental area).

Collectively, the effects of phospholipids on anxiety-like behaviors differed between different FA species at *sn*-2. Our findings indicate that a vinyl–ether bond at *sn*-1 has anxiolytic functions if AA is bound to *sn*-2, whereas phosphoethanolamine and an ether linkage contribute to reducing anxiety if DHA is bound to *sn*-2 (Figure 10B).

### 4.3. Involvement of Phospholipid Moieties in Social Interaction and Recognition Memory

Our results suggest that palmitic acid(16:0) at *sn*-1 inhibits social interaction in PC and PE with DHA at *sn*-2 (Figure 10C). As well as affecting exploratory and anxiety-like behaviors, DHA at *sn*-2 in PE had a different effect from AA as follows: DHA may be involved in the inhibition of social interaction, whereas AA appears to maintain social interactions (Figure 10C). Additionally, in trial 3 of the social interaction test, the higher discrimination index of rats injected with PC(O-18:0/22:6) than of rats injected with PC(16:0/22:6) but not PC (O-16:0/22:6) or PE(O-16:0/22:6) suggests that an alkyl–ether bond and stearic acid at *sn*-1 combine to enhance sociability with a familiar rat (Figure 10C). By contrast, neither social interaction with a novel intruder in trials 2 and 4 nor novel recognition memory of objects was affected by PC(O-18:0/22:6) injection. The stimulation of supramammillary nucleus terminals in the hippocampal CA2 reduces social memory expression and enhances social interaction, whereas the supramammillary nucleus–dentate gyrus pathway is associated with contextual novelty [36]. In our tracer study, fluorescent PE was detected in the hippocampus, including the CA2; PC(O-18:0/22:6) may therefore enhance sociability with a familiar rat by activating the supramammillary nucleus–CA2 pathway. Furthermore, projections from the ventral tegmental area to the nucleus accumbens modulate social behavior but not novel object interaction via type 1 dopamine receptor signaling in the nucleus accumbens [37]. Additionally, glutaminergic neuronal activity in the posterior intralaminar complex of the thalamus correlates to social investigation [38]. Moreover, interrupting contralateral basolateral amygdala connectivity impairs social interaction, and a mouse model of autism spectrum disorder that lacked the posterior part of the anterior commissure (*Tbr1*^+/−^ mice) showed sociability deficits [39,40]. In the present study, decreased social interactions without any changes in recognition memory suggest that the hippocampal CA2, ventral tegmental area, posterior intralaminar complex, and basolateral amygdala neurons (or the myelin surrounding their axons) are the possible targets of PC(16:0/22:6), PC(O-16:0/22:6), PE(O-16:0/22:6), PE(18:0/18:0), and PE(18:0/22:6) (because fluorescence from injected liposomes was detected in these brain areas). Similarly, matrix-assisted laser desorption/ionization mass spectrometry imaging revealed that, compared with control mice, PE ions with DHA at *sn*-2—including PE(18:0/22:6)—had higher intensities in the cortex, hippocampus, thalamus, hypothalamus, basal ganglia, midbrain, pons, and cerebellum of mice in a valproic-acid-induced autistic mouse model that exhibited social deficits [27]. By contrast, PC(O-18:0/14:0) ions, which have an alkyl–ether bond and stearic acid at *sn*-1, had lower intensities in the hippocampus and basal ganglia in this mouse model compared with controls [27]. These previous findings support our results that DHA at *sn*-2 inhibits social interaction, whereas a combination of an alkyl–ether bond and stearic acid at *sn*-1 enhances sociability. Thus, chemical structures at *sn*-1 and *sn*-2 may be the key to controlling sociability.

## 5. Conclusions, Limitations, and Future Directions

Our results suggest that the effects of phospholipids on exploration, anxiety-like behaviors, and sociability in male rats can be modulated by a single or combinatorial effect of the moiety (i.e., the chemical bond at *sn*-1, the FAs at *sn*-1 and *sn*-2, and/or the hydrophilic moiety at *sn*-3). However, there are also some limitations to our study. First, there were a lack of data involving the dynamics, metabolisms, and target cells of the injected phospholipids. Second, the physicochemical interactions between phospholipid moieties and biomolecules remain relatively unknown. Third, more phospholipids composed of different moieties need to be examined to further confirm the functions of the phospholipid moieties that were suggested in our study. Notably, previous studies have indicated that phospholipid compositions in brain regions such as the prefrontal cortex are altered in psychological (e.g., schizophrenia), neurodevelopmental (e.g., autism spectrum disorder), and neurodegenerative (e.g., Alzheimer’s disease) disorders [41,42]. Furthermore, neuropsychological disturbances can be relieved by the incorporation of highly functional phospholipids into the brain using liposomes, as described in the present study. Thus, to develop sophisticated phospholipids for treating neurodevelopmental and mental disorders, further studies elucidating the associations between the behavioral effects of phospholipids and the chemical structures of their moieties are warranted.

## Figures and Tables

**Figure 1 pharmaceutics-16-00762-f001:**
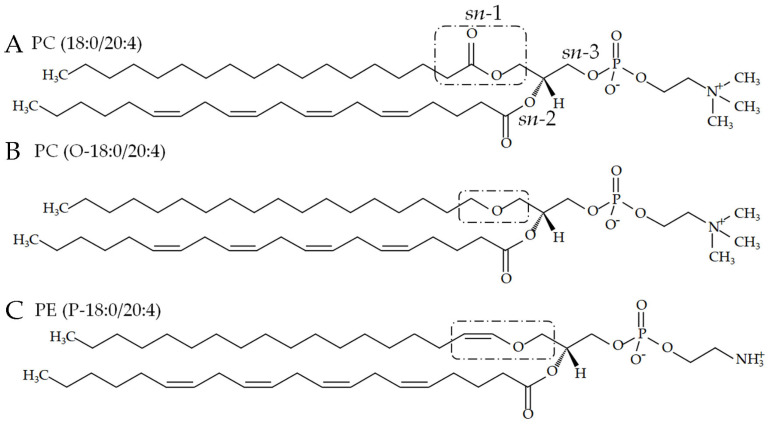
Chemical structures of phospholipids with ester, alkyl–ether, and vinyl–ether bonds at *sn*-1. The dotted boxes indicate (**A**) PC(18:0/20:4) with an ester bond, (**B**) PC(O-18:0/20:4) with an alkyl–ether bond, and (**C**) PE(P-18:0/20:4) with a vinyl–ether bond. FAs are linked to *sn*-1 and *sn*-2, and phosphocholine or phosphoethanolamine is linked to *sn*-3.

**Figure 2 pharmaceutics-16-00762-f002:**
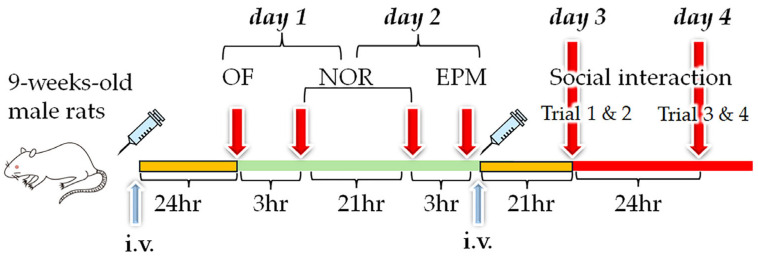
Experimental schedule. Liposomes were injected 21 h before the open-field (OF) test and after the elevated plus maze (EPM) test on day 2 (i.e., 21 h before the social interaction tests on day 3). The OF test was conducted on day 1, followed by the novel object recognition (NOR) test (days 1 and 2), EPM test, and trials 1 and 2 (day 3) and 3 and 4 of the social interaction test (day 4) [18].

**Figure 3 pharmaceutics-16-00762-f003:**
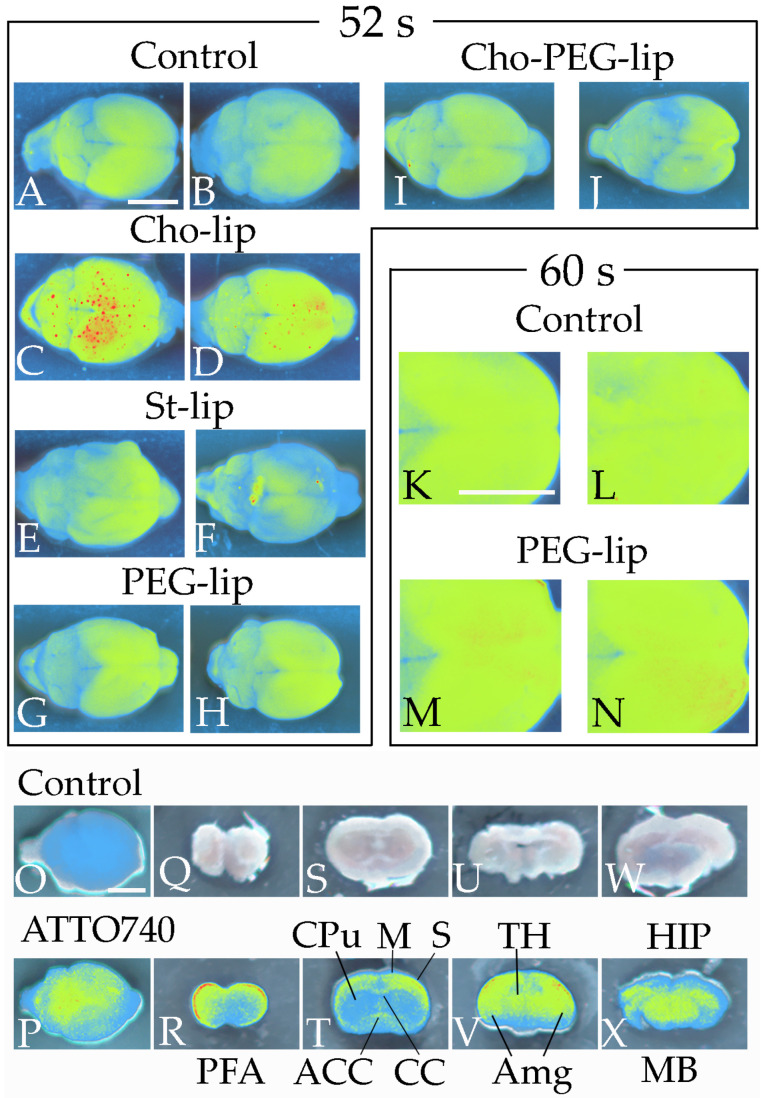
DiIC_18_ and ATTO 740 DOPE, incorporated into liposomes, accumulate in specific brain areas as revealed by in vivo fluorescence imaging. Fluorescence images of the dorsal side of the brain were acquired 3 h after control or fluorescent liposome injection (**A**–**N**). Compared with mice injected with control liposomes (**A**,**B**), strong fluorescence (red) was observed in the cerebrum and midbrain of mice injected with Cho-lip (**C**,**D**) at 52 s of exposure, but not in mice injected with St-lip (**E**,**F**), PEG-lip (**G**,**H**), or Cho-PEG-lip (**I**,**J**). At 60 s of exposure (**K**–**N**), fluorescence in the cerebrum was stronger in mice who received liposomes containing PEG than in mice injected with control liposomes. Similar to DiIC_18_, fluorescence was detected in the cerebrum and midbrain after the injection of liposomes with ATTO 740 DOPE (**O**,**P**). In the coronal brain slices (**Q**–**X**), ATTO 740 DOPE accumulated in the prefrontal area (PFA), motor cortex (M), somatosensory cortex (S), thalamus (TH), amygdala (Amg), hippocampus (HIP), midbrain (MB), anterior commissure (ACC), and corpus callosum (CC), but not in the caudate putamen (CPu). Scale bar: 5 mm.

**Figure 4 pharmaceutics-16-00762-f004:**
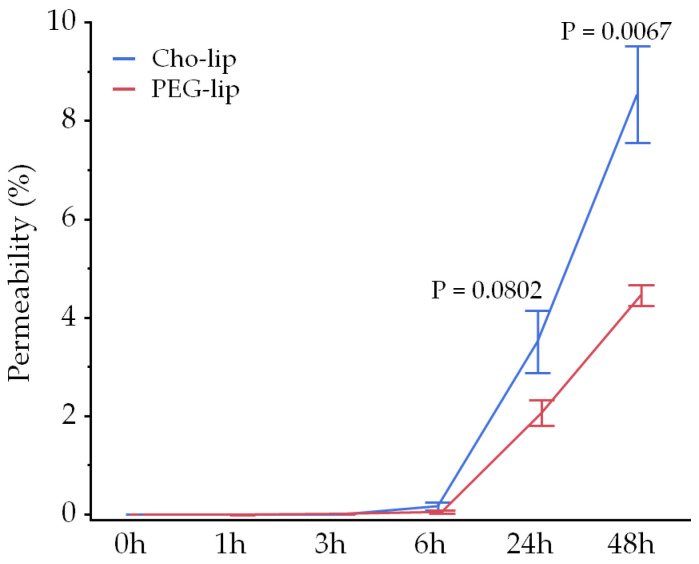
Liposomes containing cholesterol cross the BBB model more than liposomes containing PEG. Fluorescence of DiIC_18_ was barely detected in the BBB model within 3 h after adding liposomes to the blood side; however, liposomes with cholesterol (Cho-lip, *n* = 4) tended to pass through the BBB more than liposomes with PEG (PEG-lip, *n* = 4) at 24 h (*p* = 0.082) and showed significantly higher permeability at 48 h compared with PEG-lip (*p* = 0.0067). Values are presented as the mean ± standard error.

**Figure 5 pharmaceutics-16-00762-f005:**
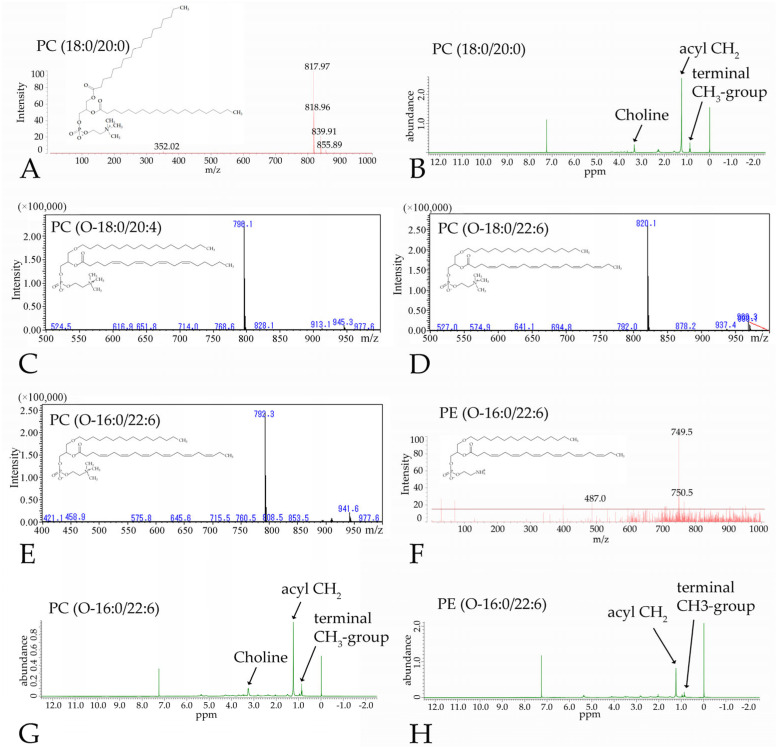
Synthetic phospholipids were identified using MALDI-TOF MS, LC-MS, and NMR. The mass spectra of PC(18:0/20:0) showed the correct mass at 817.97 *m*/*z* (**A**), and the choline peak was observed in the ^1^H NMR spectra (**B**). The LC-MS spectra of PC(O-18:0/20:4) at 798.1 *m*/*z*, PC(O-18:0/22:6) at 820.1 *m*/*z*, and PC(O-16:0/22:6) at 792.3 *m*/*z* showed peaks at the correct masses (i.e., 798.1 *m*/*z*, 820.1 *m*/*z*, and 792.3 *m*/*z*, respectively) (**C**–**E**). MALDI-TOF MS spectra of PE(O-16:0/22:6) showed a peak at the correct mass at 749.5 *m*/*z* (**F**). Although the choline peak in the NMR spectrum was observed in PC(O-16:0/22:6) (**G**), no choline peak was observable in PE(O-16:0/22:6) (**H**).

**Figure 6 pharmaceutics-16-00762-f006:**
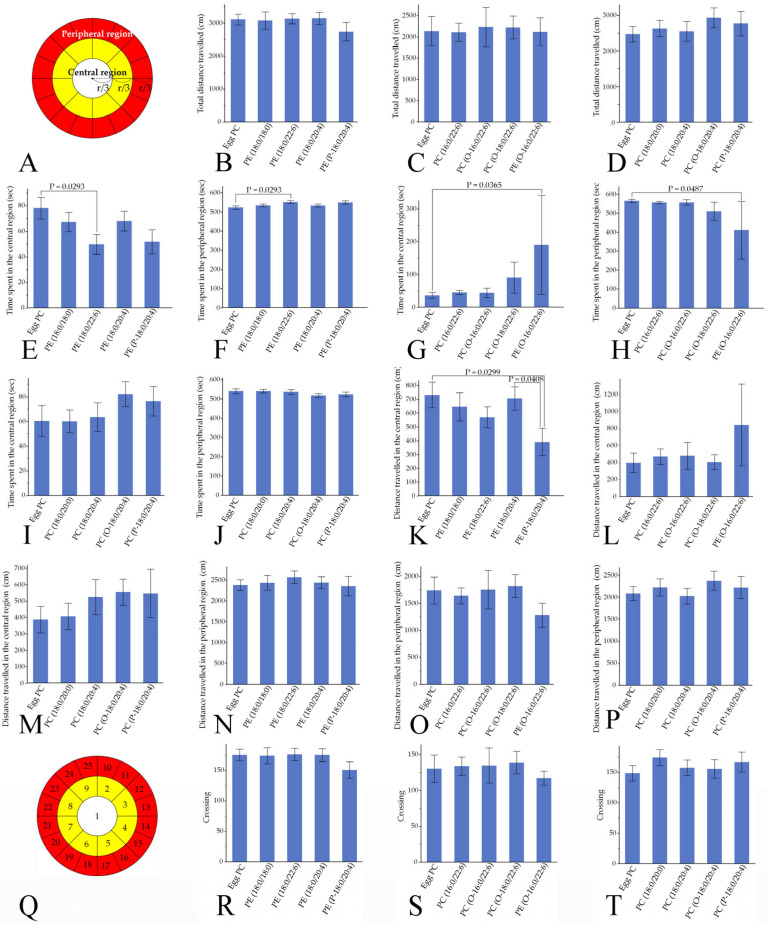
Several types of PC and PE incorporated into liposomes led to altered spontaneous activity in the OF test. (**A**) The radius of the smallest circle was one-third that of the largest circle (r), and the radius of the medium-sized circle was two-thirds that of the largest circle. The white and yellow areas show the central region, and the peripheral region is indicated as the red area. (**B**–**D**) Total distance traveled. (**E**,**G**,**I**) Time spent in the central region. (**F**,**H**,**J**) Time spent in the peripheral region. (**K**–**M**) Distance traveled in the central region. (**N**–**P**) Distance traveled in the peripheral region. (**Q**) The OF was divided into 25 areas to count the crossings. The yellow region was divided into eight equal areas, and the red region was divided into 16 equal areas. (**R**–**T**) Number of area crossings. Values are presented as the mean ± standard error. The rat numbers in each group were as follows: (**B**,**E**,**F**,**K**,**N**,**R**) Egg PC (*n* = 9), PE(18:0/18:0) (*n* = 10), PE(18:0/22:6) (*n* = 9), PE(18:0/20:4) (*n* = 10), and PE(P-18:0/20:4) (*n* = 7). (**C**,**G**,**H**,**L**,**O**,**S**) Egg PC (*n* = 10), PC(16:0/22:6) (*n* = 10), PC(O-16:0/22:6) (*n* = 6), PC(O-18:0/22:6) (*n* = 10), and PE(O-16:0/22:6) (*n* = 3). (**D**,**I**,**J**,**M**,**P**,**T**) Egg PC (*n* = 9), PC(18:0/20:0) (*n* = 10), PC(18:0/20:4) (*n* = 10), PC(O-18:0/20:4) (*n* = 9), and PC(P-18:0/20:4) (*n* = 6). Differences were evaluated using Hsu’s MCB test.

**Figure 7 pharmaceutics-16-00762-f007:**
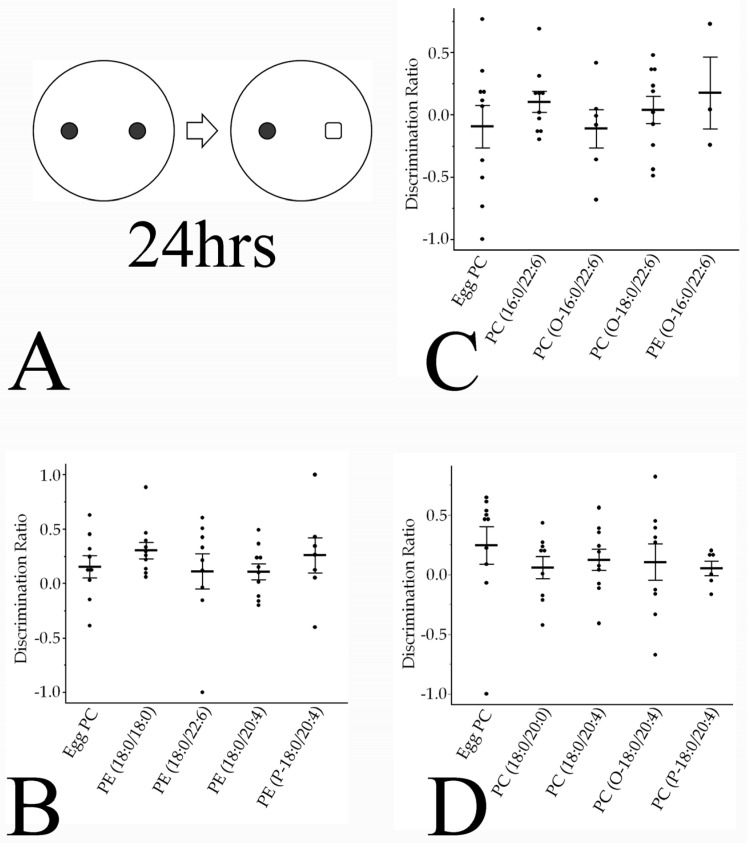
Phospholipids do not alter object recognition memory as expressed by the discrimination ratio. (**A**) Rats explored two brown cylindrical bottles placed in the arena for 10 min; 24 h later, one of the two objects was replaced with a clear square bottle filled with white molecular sieves, and the tested rats were again allowed to explore both objects for 10 min. (**B**–**D**) Values are presented as the mean ± standard error. The rat numbers in each group were as follows: (**B**) Egg PC (*n* = 9), PE(18:0/18:0) (*n* = 10), PE(18:0/22:6) (*n* = 9), PE(18:0/20:4) (*n* = 10), and PE(P-18:0/20:4) (*n* = 7). (**C**) Egg PC (*n* = 10), PC(16:0/22:6) (*n* = 10), PC(O-16:0/22:6) (*n* = 6), PC(O-18:0/22:6) (*n* = 10), and PE(O-16:0/22:6) (*n* = 3). (**D**) Egg PC (*n* = 10), PC(18:0/20:0) (*n* = 9), PC(18:0/20:4) (*n* = 10), PC(O-18:0/20:4) (*n* = 9), and PC(P-18:0/20:4) (*n* = 6). Differences were evaluated using Hsu’s MCB test.

**Figure 8 pharmaceutics-16-00762-f008:**
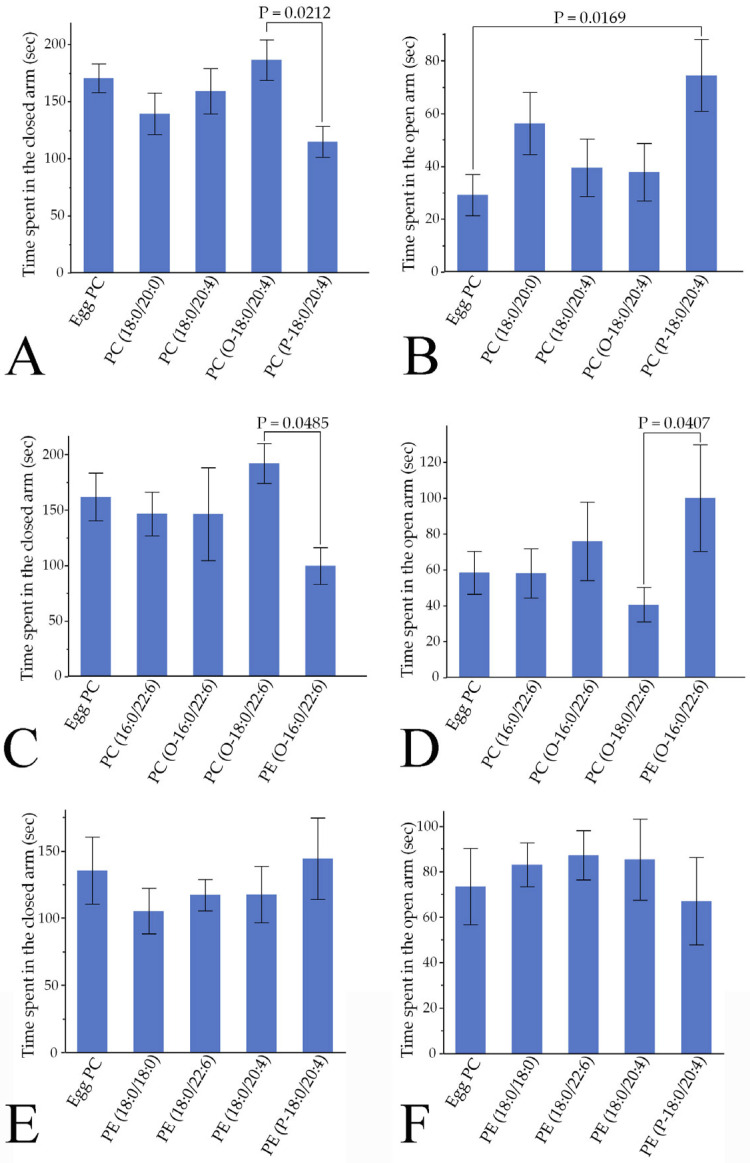
Two types of phospholipids act on anxiety-like behavior in the EPM test. (**A**,**C**,**E**) Time spent in the closed arms. (**B**,**D**,**F**) Time spent in the open arms. Values are presented as the mean ± standard error. The rat numbers in each group were as follows: (**A**,**B**) Egg PC (*n* = 9), PE(18:0/18:0) (*n* = 10), PE(18:0/22:6) (*n* = 9), PE(18:0/20:4) (*n* = 9), and PE(P-18:0/20:4) (*n* = 7). (**C**,**D**) Egg PC (*n* = 9), PC(16:0/22:6) (*n* = 10), PC(O-16:0/22:6) (*n* = 5), PC(O-18:0/22:6) (*n* = 9), and PE(O-16:0/22:6) (*n* = 3). (**E**,**F**) Egg PC (*n* = 10), PC(18:0/20:0) (*n* = 10), PC(18:0/20:4) (*n* = 10), PC(O-18:0/20:4) (*n* = 9), and PC(P-18:0/20:4) (*n* = 6). Differences were evaluated using Hsu’s MCB test.

**Figure 9 pharmaceutics-16-00762-f009:**
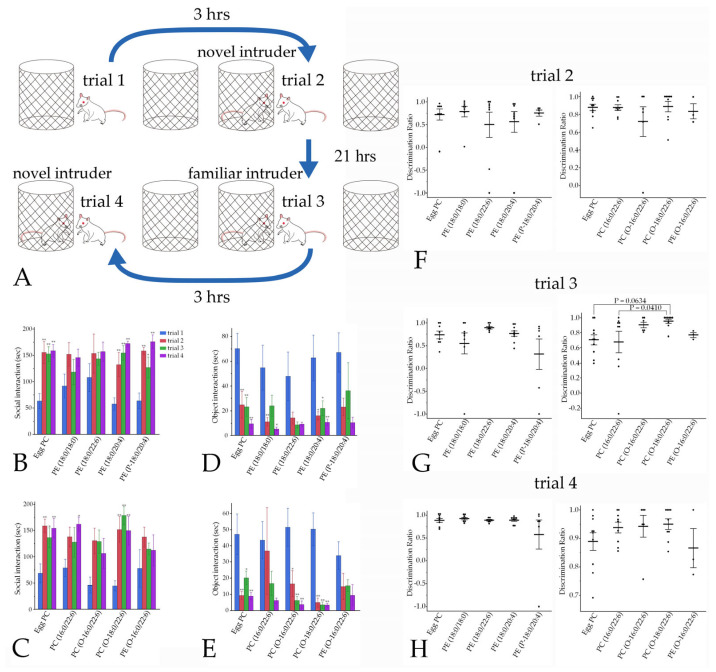
Effects of phospholipids on social interaction. (**A**) Three hours after trial 1, the social interaction time of male rats with an intruder was measured in trial 2. Rats were reintroduced in the same context as in trial 2, 21 h after the first exposure to the familiar intruder rat, in trial 3. Three hours later, male rats were tested for interaction with a novel intruder in trial 4. (**B**,**C**) Social interaction. (**D**,**E**) Object interaction. (**F**–**H**) Discrimination ratios. Values are presented as the mean ± standard error. The rat numbers in each group were as follows: (**B**,**D**) Egg PC (*n* = 8), PE(18:018:0) (*n* = 8), PE(18:0/22:6) (*n* = 8), PE(18:0/20:4) (*n* = 9), and PE(P-18:0/20:4) (*n* = 6). (**C**,**E**) Egg PC (*n* = 10), PC(16:0/22:6) (*n* = 10), PC(O-16:0/22:6) (*n* = 6), PC(O-18:0/22:6) (*n* = 9), and PE(O-16:0/22:6) (*n* = 3). Differences were evaluated using the Tukey–Kramer HSD test (**B**–**E**) and Hsu’s MCB test (**F**–**H**). * *p* < 0.05, ** *p* < 0.01.

**Figure 10 pharmaceutics-16-00762-f010:**
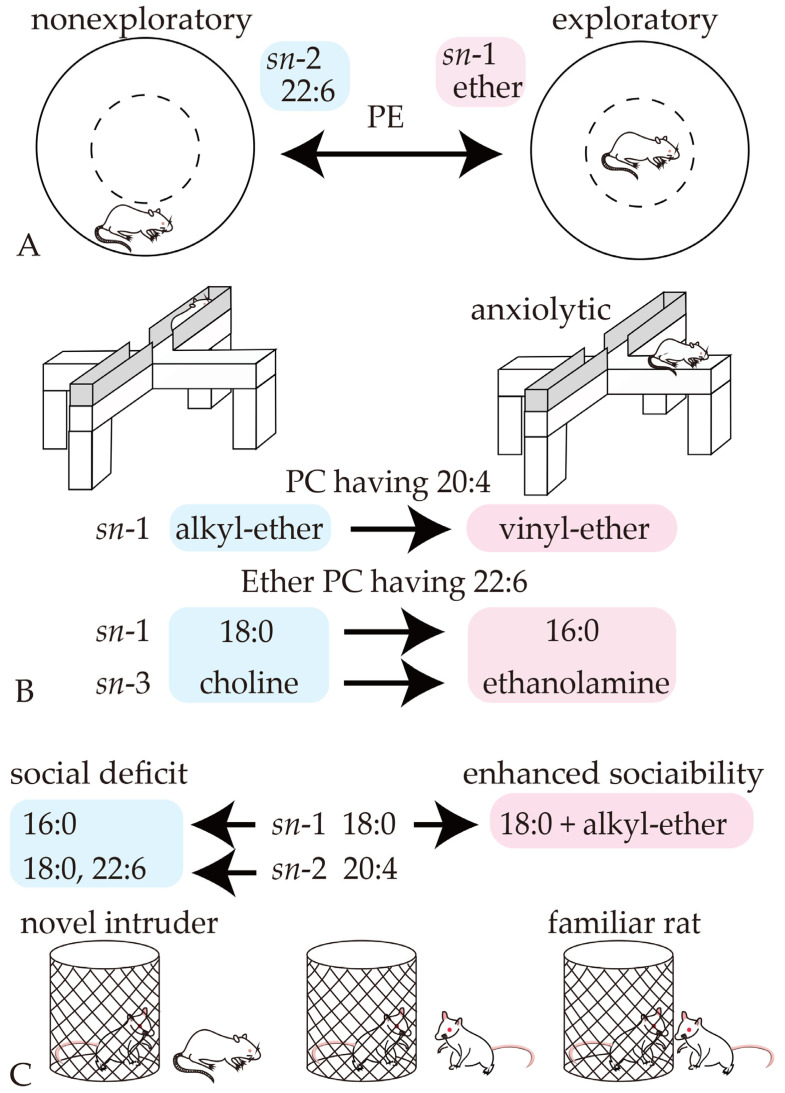
The impact of phospholipid moiety substitutions on behaviors. (**A**) DHA at *sn*-2 and an ether linkage at *sn*-1 in PE may control exploratory behaviors. (**B**) A vinyl–ether linkage at *sn*-1 in PC with 20:4, and palmitic acid at *sn*-1 and the substitution of choline with ethanolamine at *sn*-3 in PC with 22:6, may reduce anxiety-like behaviors. (**C**) Palmitic acid at *sn*-1 and stearic acid and DHA at *sn*-2 may reduce sociability. By contrast, alkyl–ether linkage and stearic acid at *sn*-1 may combine to enhance social interaction with a familiar rat.

**Table 1 pharmaceutics-16-00762-t001:** Composition of the liposomes.

Liposomes	Egg PC (mg/mL)	DSPE (mg/mL)	DiIC18 (mg/mL)	Cholesterol (mg/mL)	Methyl PEG2000-DSPE (mg/mL)
St-lip	8	2	100	0	0
Cho-lip	8	2	100	1	0
PEG-lip	8	1.5	100	0	0.5
Cho-PEG-lip	8	1.5	100	1	0.5
Control liposomes	10	0	0	0	0

DSPE: 1,2-Distearoyl-*sn*-glycero-3-phosphoethanolamine. PEG: polyethylene glycol. DSPE and Methyl PEG2000-DSPE were purchased from TCI.

## Data Availability

Data will be available upon request.

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
