# Peer review of "Solitary and Synergistic Effects of Different Hydrophilic and Hydrophobic Phospholipid Moieties on Rat Behaviors"

_pharmaceutics, 2024, doi:10.3390/pharmaceutics16060762_

Round 1

Reviewer 1 Report

Comments and Suggestions for Authors

In this manuscript Kikuchi et al. detailed the obtaining and characterization of different hydrophilic and hy- 2 drophobic phospholipid moieties, and the assessing of their effects on some behavioral experimental models in rats.

After reading the manuscript, the following doubts and suggestions have arisen.

The abstract must be redone, clearly stating the design of the study, the results and the synthetic conclusions.

At the end of the introduction section, the objective of the work and how it was carried out, should be briefly mentioned. The methodology of the research should be more detailed.

In the discussion section, authors are encouraged to thoroughly expand upon their findings by constructing well-supported arguments based on current understanding and acquired results. This entails examining potential implications, limitations, weaknesses, and avenues for future research, among other relevant factors. Authors should aim to clarify both the theoretical importance and practical implications of their research, emphasizing its relevance to real-world application.

Some other aspects were found in this manuscript:

- different fonts were used in the text and in the figures;

- the references should be upgraded;

- the references should be modified according to the journal`s guide, using the abbreviated journal name;

- a schematic representation of the study would be appreciated;

- spelling check of the text is mandatory;

Comments on the Quality of English Language

Enhancing English proficiency, including grammar, style, and syntax, should be pursued through professional guidance provided by a specialized English Editing Company that focuses on scientific writing.

Reviewer 2 Report

Comments and Suggestions for Authors

The manuscript from Kikuchi and Cols claims the differences in behavioral effects of phospholipid moieties on exploratory,  anxiety, and social behaviors

The experimental design is appropriate, and the results support its conclusion; however, the discussion lacks a plausible mechanism that explains the differences in the results. Otherwise, the manuscript is clear and can be published. 

Reviewer 3 Report

Comments and Suggestions for Authors

The manuscript entitled “Solitary and synergistic effects of different hydrophilic and hydrophobic phospholipid moieties on rat behaviors" by Kikuchi et al is potentially an important study, performed and connected finely. The study has addressed the different types of hydrophilic and hydrophobic phospholipid moieties on rat behaviors.

Following concerns need to be addressed and reconciled which could improve/upgrade this manuscript.

  1. In “liposome injections for behavior tests” authors should write about the vehicle and injection time.
  2. References are required for the open field test.
  3. For behavioral testing, authors should indicate the timing of behavioral assessment.
  4. Author needs to add the references for liposome injections protocol.
  5. If there is any supplementary data regarding characterization study of zeta potential, SEM, TEM they have to include it.
  6. Has the author checked how long the effect lasts after injection?

Thanks

Round 2

Reviewer 1 Report

Comments and Suggestions for Authors

The authors have significantly revised the manuscript addressing the concern raised. I consider it could be accepted for publication in this journal.